# Systematic Review of Therapeutic Physical Exercise in Patients with Amyotrophic Lateral Sclerosis over Time

**DOI:** 10.3390/ijerph18031074

**Published:** 2021-01-26

**Authors:** Laura Ortega-Hombrados, Guadalupe Molina-Torres, Alejandro Galán-Mercant, Eduardo Sánchez-Guerrero, Manuel González-Sánchez, María Ruiz-Muñoz

**Affiliations:** 1Department of Physiotherapy, Faculty of Health Sciences, University of Málaga, 29071 Málaga, Spain; lauraortegah@outlook.es (L.O.-H.); esanchezg@uma.es (E.S.-G.); 2Department of Nursing Sciences, Physiotherapy and Medicine, Faculty of Health Sciences, University of Almería, 04120 Almería, Spain; guada.lupe@ual.es; 3MOVE-IT Research Group, INIBICA Institute, University of Cádiz, 11110 Cádiz, Spain; 4Department of Nursing and Physiotherapy, University of Cádiz, 11003 Cádiz, Spain; 5Biomedical Research and Innovation Institute of Cádiz (INIBICA) Research Unit, Puerta del Mar University Hospital, University of Cádiz, 11003 Cádiz, Spain; 6Institute of Biomedicine of Málaga (IBIMA), 29010 Málaga, Spain; marumu@uma.es; 7Department of Nursing and Podiatry, Faculty of Health Sciences, University of Málaga, 29071 Málaga, Spain

**Keywords:** amyotrophic lateral sclerosis, physical therapy, rehabilitation, exercise

## Abstract

Background: the main objective of this study was to analyze the potential short-, medium- and long-term effects of a therapeutic physical exercise (TFE) programme on the functionality of amyotrophic lateral sclerosis (ALS) patients, measured with the Revised Amyotrophic Lateral Sclerosis Functional Scale (ALSFRS-R) scale. Methods: a systematic review of the PubMed, SCOPUS, Cochrane, Scientific Electronic Library Online (Scielo), Physiotherapy Evidence Database (PEDro), Cumulative Index of Nursing and Allied Health Literature (CINAHL) and Medical Literature Analysis and Retrieval System Online (MEDline) databases was carried out. The information was filtered using the following Medical Subjects Heading (MeSH) terms: “Amyotrophic lateral sclerosis”, “Physical Therapy”, and “Physical and Rehabilitation Medicine”. The internal validity of the selected documents was evaluated using the PEDro scale. The study included clinical trials published in the last 5 years in which one of the interventions was therapeutic physical exercise in patients with ALS, using the ALSFRS-R as the main outcome variable and functional variables as secondary variables. Results: 10 clinical trials were analyzed, with an internal validity of 5–7 points. The TFE groups showed significant short-, medium- and long-term differences, obtaining a mean difference of 5.8 points compared to the 7.6 points obtained by the control groups, at six months, measured with ALSFRS-R. In addition, the participants showed significant improvements in functional abilities in the short, medium and long terms. Conclusions: Therapeutic physical exercise could contribute to slowing down the deterioration of the musculature of patients with ALS, thus facilitating their performance in activities of daily living, based on the significant differences shown by these individuals in the short, medium and long term both in subjective perception, measured with ALSFRS-R, and functional capacities.

## 1. Introduction

Amyotrophic lateral sclerosis (ALS) is a disease of the central nervous system (CNS) and is characterized by a progressive involution of motor neurons in the cortex of the brain (upper motor neurons) [1]. As a consequence, muscle weakness occurs, which causes paralysis, spreading from different body regions. It compromises motor autonomy, written and oral communication, swallowing and breathing; however, the ocular muscles, sensitivity and intellect are not altered [2]. The clinical manifestations of ALS are due to the abnormal behavior of the nervous system [3].

The treatment of this disease must be comprehensive and must be approached from a multidisciplinary point of view, from the moment it is diagnosed until its terminal phase. It includes pharmacological, neurorehabilitation and symptomatic treatments [4]. Physical therapy consists in planned therapeutic physical exercise to correct postural abnormalities, combat pain and reduce muscle stiffness. In addition, it promotes functional independence, trains the patient to prevent falls, and re-educates walking, with certain technical aids if necessary [5]. The role of physical exercise begins prior to significant loss of strength and continues throughout the course of the disease, up to the last days of the person’s life.

Everything indicates that exercise can be physically and psychologically important for people with ALS; however, although recent studies focus on what type of exercise is most indicated for these patients, there is still no evidence of this or at what frequency or intensity should exercise be performed in training sessions or to what extent it helps patients maintain functionality.

On the other hand, the process of evaluating the patient is very important in order to analyze the effects of the intervention that is being used. In this sense, having a tool that allows studying the status and evolution of the patient with ALS is very important [6,7]: The Revised Amyotrophic Lateral Sclerosis Functional Scale (ALSFRS-R) is a measure used to assess the status and progression of patients with ALS. It is one of the most widely used scales to assess the functionality of these patients. The main components of this scale are [4]: bulbar function: language, salivation and swallowing; fine motor: writing, use of cutlery, dressing and personal hygiene; gross motricity: rolling over in bed, walking and climbing stairs; and respiratory function: dyspnoea, orthopnoea and respiratory insufficiency. It is common for people with ALS to obtain their diagnosis 1 year after the onset of the disease and with a score greater than 39 on the ALSFRS-R scale [7]. Using a common assessment tool in patients with ALS allows comparing the effects of the different proposed interventions. Furthermore, due to the functional impairment suffered by this type of patient, specifically analyzing their functional capacities is also relevant.

No systematic review has been found to analyze the effect of therapeutic physical exercise in patients with ALS in the short, medium and long term. Therefore, the main objective of this study was to analyze the potential short-, medium- and long-term effectiveness of a therapeutic physical exercise programme on the functionality of patients suffering from ALS, measured with the ALSFRS-R scale. The secondary objective of this systematic review was to analyze the effect of therapeutic physical exercise on functional variables in patients with ALS in the short, medium and long term.

## 2. Materials and Methods

### 2.1. Search Strategy

To carry out this systematic review, information collected from the PubMed, SCOPUS, Cochrane, Scientific Electronic Library Online (Scielo), Physiotherapy Evidence Database (PEDro), Cumulative Index of Nursing and Allied Health Literature (CINAHL) and Medical Literature Analysis and Retrieval System Online (MEDline) databases was used. For the selection of the information, descriptors obtained from Medical Subjects Heading (MeSH) were used. The information was filtered using the following keywords: “Amyotrophic lateral sclerosis”, “ALS”, “Motor Neuron Disease”, “Physical Therapy”, “Exercise Therapy”, “Physical and Rehabilitation Medicine”, and “Exercise Training”. A systematic review of the scientific literature was carried out following the Preferred Reporting Items for Systematic Reviews and Meta-Analyses (PRISMA) model (Figure 1). The search was carried out between 15 and 30 June 2020. The PICOT (Patient; Intervention; Comparison; Outcome; Time) methodology was followed to evaluate the effect of therapeutic physical exercise, in the short, medium and long term, in patients with ALS, compared with control groups and evaluated with the ALSFRS tool-R, as the main outcome variable, and with functional tests, as secondary variables.

### 2.2. Selection Method

Two researchers with more than 10 years of experience in the selection of documents performed the blind selection of the different documents. After applying all the previously described criteria, those that were duplicated were discarded and the selected articles were read in full by the authors. In the event of a discrepancy in any criteria, a third researcher (blinded) with more than 15 years of experience in document selection, decided if the paper should be included.

### 2.3. Selection of Documents

The following inclusion criteria were established: clinical trials published in the last 5 years, with at least one of the interventions being therapeutic physical exercise in patients with ALS, which included the values of the ALSFRS-R as an outcome variable of patients who underwent study, in addition to obtaining a score ≥5 on the PEDro methodological quality scale.

The following exclusion criteria were assigned: score <5 on the PEDro methodological quality scale; the absence of the outcome variable ALSFRS-R among the measurement values; trials with exclusively respiratory/bulbar rehabilitation programmes; or trials involving animals. Furthermore, studies that were not published in English, Spanish, French, Portuguese or Italian were excluded.

### 2.4. Evaluation of the Internal Validity of the Selected Documents

The PEDro methodological assessment scale was applied to estimate the quality of the analyzed studies [8]. This scale consists of 10 items plus selection criteria: (1) randomization of the sample; (2) concealed allocation; (3) initial comparability between groups; (4) all subjects blinded; (5) all therapists who administer therapy blinded; (6) all evaluators measuring key outcomes blinded; (7) adequacy of follow-up; (8) analysis with intention to treat; (9) statistical comparison of results between groups; and (10) existence of specific measures and variability for at least one key result. These items are dichotomous (i.e., 1 point per item), thus points were obtained based on compliance with the requirements of each particular item [8].

Studies with a score ≥6 were considered high-quality studies.

## 3. Results

Initially, 753 documents were found in the PubMed, SCOPUS, Cochrane, SciELO, PEDro, CINAHL and MEDLINE databases. The following were excluded: 467 duplicates, 155 older than 5 years, 84 non-clinical trials, 11 animal studies, 9 that only contained bulbar rehabilitation and 17 that did not include the main outcome variable of the ALSFRS-R. After this, 10 clinical trials were analyzed in full-text and included (Figure 1).

The 10 selected clinical trials were evaluated with the PEDro scale. The score ranged from 5 to 7 points. Four of the studies obtained 5 points, three of them obtained 6 points and the rest obtained 7 points. Furthermore, none of them were masked, and they all included the results of the primary variable (Table 1).

The characteristics of all the documents were analyzed (Table 2). As can be seen, a total of 421 patients were test subjects, of which 183 underwent rehabilitation with physical exercise and were part of the group of cases; the rest of the participants belonged to the control group and their treatment was mostly passive. The mean age of the patients was 60 years, and the mean time they suffered from the disease was 15 months.

Some of the therapeutic physical exercise interventions consisted of aerobic exercise [9,10], moderate-high-intensity strength and endurance exercises [5,11,12,13], functional training or stretching [14,15]. Frequencies ranged from highest to lowest [10] according to the study, and the intervention time was a minimum of 2 weeks and a maximum of 6 months. The exercises were performed at around 70% heart rate (HR), and the strength of each patient was measured in order for them to avoid exceeding 80% of their MR.

The outcome variables of each article were analyzed. The main one was the ALSFRS-R functionality scale (Table 3), in which the score ranged between 32 and 43 points for the initial minimum and maximum score, respectively. In addition, these documents included secondary variables, such as forced vital capacity (FVC), fatigue severity scale (FSS) and 6 min walking test (6MWT). The scores were collected at the start of the treatment and in the short, medium and long term. They were divided into two tables: ALSFRS-R (all studies) (Table 3) and FVC, FSS, and 6MWT (7/10 studies were included) (Table 4). We chose these variables according to their relevance and the frequency with which they occurred in the analyzed studies. Both tables show the evolutionary comparison of cases and controls, in addition to the standard deviation of each data unit.

## 4. Discussion

This study was focused on analysing the functional changes that a therapeutic physical exercise programme can cause in patients suffering from ALS as measured by the ALSFRS-R scale. After a search and exclusion of clinical trials in different databases (Figure 1), 10 of these studies were analyzed in full-text, which were evaluated using the PEDro scale (Table 1). In them, 421 patients were analyzed between groups of cases (treated with therapeutic exercise) and control groups (with habitual passive treatments), whose results were obtained with different scales, mainly the ALSFRS-R (a scale of functionality that is currently accepted to quantify the capacity of these patients to carry out their routine activities [4]), and FSS, FVC and 6MWT were established as secondary measures, based on their frequency of appearance and correlation with the study. Furthermore, due to the different duration of each clinical trial, it was possible to determine the effect of neurorehabilitation at different stages of the disease. Thus, the effect of the intervention can be compared in the short, medium and long term and behavioral trends can be observed between the scores of the different patients treated. Likewise, the following results were obtained from them (Table 3).

In most of the clinical trials in which the effect of a treatment based on physical exercise is studied in the progression of ALS, the strength of the patients is usually improved in the trained muscles. However, there is still discrepancy regarding functionality. In fact, there are claims about possible ineffectiveness in improving function in patients with ALS through moderate- to high-intensity strength exercise therapy according to the Borg Scale [18]. In this sense, it was found that, by training the lower limbs of the patients who participated, they managed to increase strength in actions such as knee extension, although with no progress in activities such as climbing stairs or walking. Faced with this question, the results of the analyzed studies are broken down according to the moment.

### 4.1. Short-Term Effects

In the first month of rehabilitation, even improvements in the ALSFRS-R score can be found [16]. This is a relevant fact, since, when dealing with the rehabilitation of a degenerative disease, function is expected to be lost with the passage of time [3]. In fact, in the rest of the trials in which measurements were taken after 1 month, it was found that the scores of cases and controls decreased around 2 points [15] or remained the same [11]. This is considered a positive aspect of progress and, in view of these qualifications, it seems that therapeutic physical exercise manages to slightly improve (+0.5 points [16]), maintain [11] or slow down the natural degenerative progress of the patients (−1.8 points [15]), as observed when comparing the control groups of the respective selected trials.

Comparing the results of one of the groups with a study that carried out a follow-up at 1 month, a significantly greater reduction in the score can be observed [15]. This could be due to the frequency of intervention, since their groups performed physical exercise only twice per week, which is a more reduced continuity than in the rest of the studies. In addition, this trial included patients with a disease duration of <2 years and a very high ALSFRS-R score (39.1 points). This fact did not occur in the rest of the patients and it may indicate that the behavior of the pathology, which decreased at 1 month, could return to the normal level of affectation in the medium term, since it is difficult to maintain such high score levels in a disease that is characterized by the early degeneration of nervous tissue [19].

It seems that, after a month of rehabilitation, the effects of physical exercise are still not clear and that the hypertrophy caused by physical rehabilitation appears after one month [18]. Prior to this, the so-called recruitment by collateral outbreak takes place, which is caused by muscle cells without neurological damage [20]. This could explain why there are some cases in which function is improved early in this type of rehabilitation. However, in other pathologies, such as multiple sclerosis, there are studies in which improvements in physical condition linked to quality of life have been observed in only 3 weeks of rehabilitation training, with characteristics similar to the exercise plans proposed in the studied patients with ALS—moderate intensity, twice per week and aerobic skills combined with strength exercise [21].

### 4.2. Medium-Term Effects

It seems that, within 3 months, significant differences begin to appear between the groups treated with therapeutic exercise and those treated with passive conventional treatment [12,15,17] (Table 3 and Table 4). In all of them, the ALSFRS-R scores decreased between 0.13 [16] and quasi 5 points [10,12] in the training group compared to those in the control and passive therapy groups, which had a decrease in the greater functionality score. However, a different outcome was also reported, in which a group of participants who trained less frequently (two sessions/week) obtained better results than a group that performed the exercises five times/week [10]. This fact supports the theory that training must be balanced between muscle overwork and underwork [22]. This is also correlated with the involutional factor of ALS, in which valid neurons decrease as time passes, and it is recommended not to tire the patient [18], thus recommending a treatment with moderate rather than intense tendency. In addition, a study states that mild and moderate exercises (such as a swimming regimen) help to preserve motor neurons, and that more intense levels obtain the opposite effect, even decreasing survival [23].

Furthermore, other studies have shown that there may not be significant differences in physical health status in people with chronic musculoskeletal disorders who work out twice per week compared to three times per week. This is to be expected, since, once a point of maximum tolerated load has been reached, the benefits of training no longer show significant improvements. Therefore, the costs of resources and treatments could be saved without apparently affecting these results [24].

Regardless of the intensity and frequency of the sessions throughout the week, it can be seen that the duration of therapy did not exceed 60 min per session in any of the analyzed trials. Although the effort tolerance of a patient with ALS could be lower than that of a healthy subject, this could lead to a regression in therapy and in their own disease. Therefore, this must be explored, focusing on finding a balance between the accumulated load and the frequency of sessions and their duration.

It was also observed that there were scores on the functionality scale that decreased more than in the rest of the studies that carried out the measurement at 3 months and whose cases and controls obtained very different results, since the points in the usual care control patients dropped from 35 to 23 and from 40 to 35 in the motor rehabilitation patients. However, this could be due to the fact that they started with 40 points, which is significantly higher compared to the controls [12]. Regarding the sharp decrease in this study in only 3 months in the score on the ALSFRS-R scale, the age of the patients in this trial may have had an influence compared to the rest of the analyzed studies, since the mean age in this study was 53, whereas in the other studies the mean age was 61 (Table 2). This finding is related to the fact that, as a general rule, the musculoskeletal system of a 10-year-older adult will be more impaired or have more signs of muscle immobilization than a 10-year-younger adult [25]. Therefore, in turn, this condition of weakness will show more noticeable changes and advances in patients who entered the studies with worse physical condition or, in this case, at an older age. In fact, elderly people with Parkinson’s disease underwent strength training for 4 months and were compared with a control group (passive) in terms of respiratory muscle strength and quality of life. The result was that the older people who underwent active neurorehabilitation looked significantly better [26]. For this reason, the development of this type of protocol could be of interest, since previous studies with different degenerative pathologies have shown effective results.

### 4.3. Long-Term Effects

A significant difference can be observed in most of the analyzed studies [9,14,15], with differences ranging between 3 [14] and 6.76 [10] points, with respect to the record in the baseline analysis. These values represent an involution of 6.25% and 14.1%, respectively, as patients treated with therapeutic exercise benefited most at a functional level compared to those who continued with a more sedentary lifestyle, who experienced an involution (measured with ALSFRS-R) of between 6.13 [13] and 9.6 [15], which implies an involution (measured with ALSFRS-R, 0–48 points) of 12.77% and 20%, respectively. Some functional results of groups (physical exercise or stretching) were similar; however, an analysis of the falls of patients was subsequently carried out, showing that the group that worked with strength and resistance had fewer falls compared to the group that worked with joint range of motion and stretching [13]. Both the number and frequency of falls decreased in other neurodegenerative pathologies (such as Parkinson’s disease) thanks to a programme of therapeutic physical exercise [27].

However, functional improvements were seen in some patients only at the beginning of treatment when their score was higher than 40 points. Then, they underwent rehospitalization and physical exercise after those months proved to be of little efficacy in more advanced stages, thus suggesting that some patients may not respond to this type of therapy when their scores on the ALSFRS-R scale drop below 40 points and distance themselves from the initial stages of the disease [5].

Analyzing and comparing the scores obtained in the studies that measured the ALSFRS-R scale at 6 months, the behavior between the different groups of cases, but not of controls, was similar. That is, the observed trend is for a drop of approximately 6 points on the functionality scale in this period of time, having performed moderate-intensity physical exercise regardless of whether it was focused on aerobic or strength exercise. However, in the case of controls, there were groups that, with the usual therapy, dropped to 10 points from the start [9,15] or only between 4 and 7 points [10,13,14]. Therefore, it could be inferred that, in a rehabilitation that did not include therapeutic physical exercise, the possibilities of slowing down the disease would be exposed to heterogeneous conditions and the patient’s own circumstances; however, with extra physiotherapy rehabilitation, a non-evolutionary development could be expected. This is completely unfavourable for the patient, and a controlled decrease in the ALSFRS-R score is usually associated with managing activities of daily living, probably for a longer period of time than other patients whose therapy is mostly passive.

Although most of the studies cited above and analyzed in this review obtained satisfactory results with therapies based on therapeutic physical exercise, there remain divergences in some other trials in which the progression of the disease was not stopped as much as expected. This could be due to the possibility that some of these treated patients may have exceeded the intensity of the exercise, since there seems to be a correlation between patients who performed moderate exercise and a higher density of motor neurons in the ventral horn of the spinal cord, which leads to initially slower muscle deterioration, whereas the opposite occurs if the musculature is overloaded. Even sedentary patients can show better progress than those who excessively reach fatigue [28].

In the longer term, if the training therapy exceeds 10 months, it seems that only a partial improvement is observed [5], and routine activities become more difficult, thus the ALSFRS-R score in the follow-up could plummet, as in the case of [10], where it was below 30 points. When this happens, most cells are usually poorly innervated, and it is up to the “healthy” cells to counteract this weakness. For this reason, it is important to boost healthy muscle cells in time, and physical therapy in ALS could be used preventively before the onset of weakness in gait [18]. In this way, therapeutic exercise would be recommended to be implemented before and during the onset of the disease [5].

Neurorehabilitation with physical exercise seems to help strengthen the musculature in its “disuse factor”; therefore, patients with ALS who started with physical exercise and who were previously sedentary or were hospitalized would have greater results than patients who started rehabilitation in good shape [10]. This could be one of the reasons why the analyzed studies varied in terms of results and a common improvement trend was not obtained in all the exercised groups.

Although some groups of cases achieved greater progress than their control groups in terms of functionality, the same was observed regarding survival, which was not related to the ALSFRS-R score [9]. It seems that the measures of this scale are not linked to the probability of surviving a longer time [10].

However, some aspects were correlated with this scale, such as a deceleration of muscular atrophy that is normally caused by progressive inactivity, which is responsible for the involutional character of this disease [11]. In addition, aerobic fitness and respiratory function are benefited [12,14]. It is even inferred that these two aspects could influence factors such as appetite, sleep or mood [29]. For this reason, other secondary outcome variables that appeared in the reviewed studies were analyzed, which will allow relating the functionality of the ALSFRS-R scale with other aspects such as fatigue (measured with the FSS scale), the distance covered in the 6MWT and the spirometric test of FVC (Table 4).

In the case of the FSS, different results can be observed depending on the treatment. There are cases in which patients who performed resistance and strength training at 80% of their MR worsened, thus increasing their score on the FFS after 1 month of rehabilitation, obtaining even worse scores than those of the controls of the same study, which were maintained [11]. However, when reviewing the scores obtained after 1 month of treatment, another study was favorable and reduced their FFS score [16]. This could be related to the fact that, in this last study, the rehabilitation consisted in aerobic exercise and did not include strength exercises, in addition to the fact that it was performed only in three sessions per week, while in the trial that obtained the worst results, the frequency of training was daily, i.e., seven sessions per week. This shows the importance of cardiovascular training in these patients, since the respiratory muscles in more advanced stages of the disease are severely affected [30]. Cardiovascular training would be the object of study if moderate aerobic exercise would act as an adjunct against this critical factor of ALS, in order to reduce the anguish and fatigue that this generates in patients. Although, as has been seen in the case of multiple sclerosis, the mechanisms related to fatigue and corticospinal function are altered by the abnormalities in the CNS causing these neurological diseases, the response to exercise and fatigue seems to be different from that in healthy subjects [31].

However, in both of the aforementioned studies, the distance in the 6MWT was greater after 1 month [11,16]. Regardless of the fatigue that it caused, it seems that the patients who underwent strength training and perceived greater fatigue were able to walk a greater distance compared to that obtained at the beginning of the treatment. Specifically, the difference was 71 m, and no difference was observed in the controls with usual treatment. Thus, it seems that, with passive patient care, fatigue can be controlled more safely than with strength exercises, which can lead to a feeling of exhaustion [32]. However, they seem to be more capable of performing activities of daily living and their muscles could be more prepared to carry out fundamental activities (such as walking) for a longer period of time than patients with a more sedentary lifestyle, according to the results of the compared tests.

This raises questions about the relevance of a balance between perceived fatigue and the improvements obtained with physical exercise therapy, since what is observed in this review is that the appearance of fatigue is not always an indication of regression in the disease, but a stage that sometimes extends to a few days, and rest would be recommended in them, with continued improvement [11]. For this reason, recovery from this post-training weakness is of vital importance. In fact, there are studies that focus precisely on the recovery cycles of muscle speed in relation to the number of motor units and the changes that occur in sickness [33].

In the longer term, specifically 6 months after beginning of rehabilitation, the FSS was performed on patients and a score increase on this scale (which was not positive) was observed in the group that performed the tests. The group that performed exercises with the highest frequency (5 days/week) showed worse results compared to the group that had workout sessions 2 days/week [10], which is in line with the results analyzed after only 1 month of treatment. Consequently, establishing more than 3 days of weekly training could increase the exhaustion of the patients, according to the scores obtained from the FSS. If this fact is related to the measurement of FVC that this trial recorded from its patients (also at 6 months), it can be seen that the group that worked with a lower volume of sessions exceeded by more than 10% the group of cases in spirometry. Therefore, it is suggested that perhaps submaximal aerobic activity in patients with neuromuscular diseases, allowing for adequate rest, prevents overexertion [34], and that this is not effective for the achievement of greater lung capacity, although all treatments or physical exercise programmes must be adapted to the capacities of the patient with ALS [35].

Regarding the rest of the FVC percentages extracted from the long-term studies reviewed (6 months), it was observed that the patients in the control group had a decrease of around 15–25% in their spirometric values, while patients in the case groups had a decrease of around 7–15% in their spirometric values [5,13,15,17]. Therefore, it could be said that physical exercise therapy has a positive impact on slowing down the deterioration of the patient’s respiratory muscles.

### 4.4. Strengths and Weaknesses

As a strength, this is the first scientific document to study the relationship between therapeutic physical exercise in patients with ALS and the effect it produces in these subjects, both at a functional level on subjective scales (ALSFRS-R and FSS) and on objective physiological aspects (FVC and 6MWT).

In this study, the lack of revision of those articles that were not written in one of the following languages can be considered as a weak point: English, French, German, Spanish or Portuguese. Likewise, it is possible that certain nuances of the original version of the analyzed articles may not have been accurately interpreted in their translation.

In addition, those documents that were not included in the analyzed databases (PubMed, SCOPUS, Cochrane, SciELO, PEDro, CINAHL and MEDLINE) remained outside the scope of this study.

## 5. Conclusions

Therapeutic physical exercise could help to slow down the deterioration of the musculature of people with ALS, thus facilitating their performance of activities of daily living and, therefore, maintaining the levels of their scores on the ALSFRS-R functionality scale, especially in the medium and long terms, compared to those in ALS patients with mostly passive treatments.

The survival of patients did not improve in relation to a therapeutic physical exercise programme, although their respiratory muscles did improve the levels measured in the FVC, which leads to an improvement in their quality of life until the end of their disease.

Carrying out a treatment that includes physical activity in patients with ALS helps to counteract the muscle weakness caused by the degeneration of poorly innervated cells, as it strengthens healthy cells, and these patients obtained better results in the 6MWT after several months of rehabilitation.

Regarding the type of exercise, it could be inferred that moderate intensity and not very high frequencies (two sessions/week), combining strength and aerobic resistance, could be the best option to observe improvements in patients with ALS and prevent the onset of fatigue in these patients, since their FSS values may increase with more intense therapies.

## Figures and Tables

**Figure 1 ijerph-18-01074-f001:**
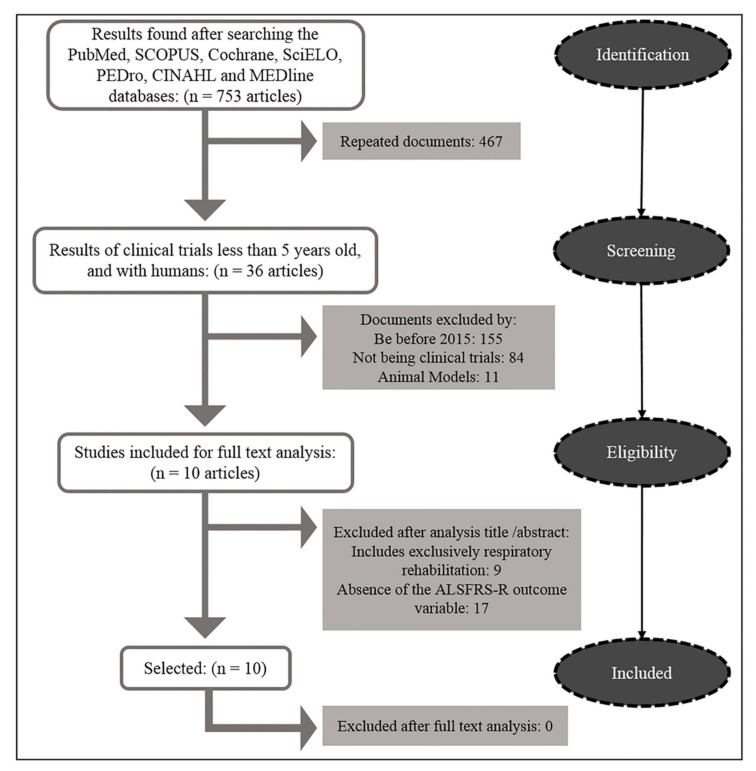
Search flow chart according to PRISMA model.

**Table 1 ijerph-18-01074-t001:** Physiotherapy Evidence Database (PEDro) scale total score of the different studies included.

Authors	Type of Study	Randomization	Masked	Variable at Startup	Blinded Subjects	Blinded Therapists	Blinded Evaluators	Measures 85% of the Sample	Intention to Treat Analysis	Main Outcome Variables	Mean or Standard Deviation	PEDro Points
Marques Braga et al., (2018) [9]	ECA	●	-	●	-	-	-	-	●	●	●	5
Kitano et al., (2018) [14]	EC	-	-	●	-	-	-	●	●	●	●	5
Sivaramakrishnan et al., (2019) [16]	EC	-	-	●	-	-	-	●	●	●	●	5
Clawson et al., (2017) [13]	ECA	●	-	-	-	-	●	●	●	●	●	6
Kato et al., (2018) [5]	EC	-	-	●	-	-	-	●	●	●	●	5
Van Groenestijn et al., (2019) [17]	ECA	●	-	●	-	-	●	●	●	●	●	7
Ferri et al., (2019) [12]	ECA	●	-	●	-	-	-	●	●	●	●	6
Merico et al., (2018) [11]	ECA	●	-	●	-	-	●	-	●	●	●	6
Zucchi et al., (2019) [10]	ECA	●	-	●	-	-	●	●	●	●	●	7
Lunetta et al., (2015) [15]	ECA	●	-	●	-	-	●	-	●	●	●	6

**Table 2 ijerph-18-01074-t002:** Characteristics of the included studies.

Author	Sample Size	Age (Years)	ALS Time	Intervention	Session Frequency	Outcome Variables
Marques Braga et al., (2018) [9]	50 (48) G1 = SC: 25 (24) G2 = AECI: 25 (24)	G1: 63 (±13.0) G2: 62 (±12.0)	G1 = SC: 9.5 months G2 = AECI: 9 months	G1 AECI: Aerobic exercise of controlled and moderate intensity + SC G2 SC: Standard care (ROM + Gear)	G1: AECI (2 sessions/week) + daily SC—6 months G2: daily SC—6 months	ALSFRS-R CPET FVC, FSS
Kitano et al., (2018) [14]	105 G1 = Home-ex: 21 (15) G2 = Control: 84	G1: 62.8 (±10.2) G2: 62.7 (±12.1)	G1: 2.2 (±2.4) years G2: 1.5 (±1.7) years	Strength, functional and stretching exercises for upper limbs and trunk muscles	Daily/individualized frequency for 6 months. G1 = unsupervised	ALSFRS-R MMT
Sivaramakrishnan et al., (2019) [16]	9 G1 = Aerobic G: 9	G1: 59.22 (±12.3)	G1: 2.37 (±1.9) years	Reclining stepped aerobic exercise of moderate intensity. 70 steps/minute.	40 min. 3 sessions a week. For 4 weeks.	ALSFRS-R 6MWD, TUG FSS, SF-12, BDI
Clawson et al., (2017) [13]	59 G1 = ROM: 21 G2 = RESISTANCE: 18 G3 = ENDURANCE: 20	G1: 57.68 (±9.72) G2: 63.65 (±10.55) G3: 57.82 (±11.88)	G1: 11.08 (±13.21) G2: 7.25 (±7.21) G3: 7.30 (±6.80) months	G1: Held stretches (30 s) G2: strength exercises (70% RM) G3: moderate/high intensity resistance exercises.	3 sessions a week. For 6 months.	ALSFRS-R FVC, FSS ASS, VAS VO2 MÁX
Kato et al., (2018) [5]	2 G1 = Strength G: 2	G1: 56 (±8)	G1: 1.1 (±0.5) years	Moderate/high intensity strength and endurance exercises.	30 min. 7 sessions a week, For 2 weeks.	ALSFRS-R KEMS, FVC FAC
Van Groenestijn et al., (2019) [17]	57 (32) G1 = AET: 27 (10) G2 = UC: 30 (22)	G1: 60.9 (±10) G2: 59.9 (±10.7)	G1: 15.5 (±10.9) G2: 18 (±14.0) months	G1 AET: Aerobic exercise therapy in cycle ergometer + UC G2 UC: usual care	50 min. 3 sessions a week. For 16 weeks.	ALSFRS-R ALSAQ-40+FS-36 MCS, PCS, FVC
Ferri et al., (2019) [12]	16 G1= TRAIN: 8 G2= UC: 8	G1: 50.7 (±3.3) G2: 55.5 (±5.95)	G1: 20.5 (±20.3) G2: 13.4 (±6.6) months	G1 TRAIN: Moderate/high intensity aerobic and strength exercise. G2 UC: usual care	50 min. 3 sessions a week. For 12 weeks.	ALSFRS-R 6MWD, TUG VO2 MAX, Mc GILL
Merico et al., (2018) [11]	38 G1 = EP: 23 G2 = SNT: 15	G1: 61.6 (±10.6) G2: 59.8 (±14.7)	G1: 30.2 (±11.8) G2: 30.3 (±6.7) months	G1 EP: Submaximal aerobic exercise 65% HR and 80% strength RM G2 SNT: neuromotor standard exercise	50 min. 7 sessions a week. For 5 weeks.	ALSFRS-R 6MWT, FIM, CK FSS, VO2 MAX
Zucchi et al., (2019) [10]	65 G1 = IER: 32 G2 = UER: 33	G1: 65.14 (±9.90) G2: 64.74 (±10.10)	G1: 15.67 (±9.74) G2: 16.64 (±8.98) months	G1: High frequency aerobic and resistance training. G2: Aerobic exercise, low frequency.	45 min. G1: 5/week and G2: 2/week For 10 weeks.	ALSFRS-R FVC FSS ALSAQ-40 + Mc GILL
Lunetta et al., (2015) [15]	60 (47) G1 = SMEP: 30 (22) G2 = UCP: 30 (25)	G1: 61.1 (±10.1) G2: 60.3 (±9.9)	G1: 15.2 (±7.2) G2: 13.7 (±6.1) months	G1 SMEP: passive, active and cycle ergometer exercises, strictly supervised. G2 UCP: passive habitual care.	G1: and G2: 2/week For 6 months.	ALSFRS-R FVC

ALSFRS-R: revised functional scale for amyotrophic lateral sclerosis, ALSAQ-40: Assessment of subjective health status in amyotrophic lateral sclerosis, ASS: Ashworth Spasticity Scale, BDI: Beck’s depression inventory, CPET: cardiopulmonary exercise test, FAC: functional walking test, FIM: functional measure of independence, FSS: fatigue severity scale, FVC: forced vital capacity, KC: creatine kinase, KEMS: strength knee extensor muscles, MCS: mental component summary, MMT: manual muscle test, PCS: physical component summary, ROM: Range of Motion; TUG: get up and walk test, VAS: visual analog scale, VO2Máx: maximum oxygen consumption, 6MWD: 6 min walk test.

**Table 3 ijerph-18-01074-t003:** Summary of the ALSFRS-R outcome variable.

Variable	Author		Baseline (SD)	Short-Term (1 Month)	Medium-Term (3 Months)	Long-Term (6 Months)
ALSFRS-R(0–48)	Marques Braga et al. (2018) [9]	Cases	40.25 (±5.00)	-	-	34.1 (±7.1)
		Control	37.25 (±4.9)	-	-	29.5 (±7.7)
	Kitano et al. (2018) [14]	Cases	41.1 (±4.5)	-	-	38.1 (±5.9)
		Control	40.3 (±4.4)	-	-	33.1 (±9.2)
	Sivaramakrishnan et al. (2019) [16]	Cases	32.75 (±7)	33.25 (±7.55)	32.62 (±7.4)	-
		Control	32.75 (±7)	-	-	-
	Clawson et al. (2017) [13]	Cases	39.36 (±4.92)	-	-	35.41 (±1.26)
		Control	39.67 (±3.71)	-	-	33.54 (±1.38)
	Kato et al. (2018) [5]	Cases	43 (±2)	-	-	37.5 (±1)
		Control	43 (±2)	-	-	-
	Van Groenestijn et al. (2019) [17]	Cases	42.4 (±4.3)	-	40.52 (±3.48)	-
		Control	42.2 (±3)	-	38.28 (±5.52)	-
	Ferri et al. (2019) [12]	Cases	40.4 (±1.5)	-	35.7 (±2.6)	-
		Control	35 (±3.4)	-	23 (±5.6)	-
	Merico et al. (2018) [11]	Cases	36.1 (±4.71)	36.1 (±4.71)	-	-
		Control	34.5 (±3.6)	34.5 (±3.6)	-	-
	Zucchi et al. (2019) [10]	Cases	39.84 (±5.7)	-	34.87 (±8.49)	33.08 (±9.76)
		Control	40.15 (±5.17)	-	36.39 (±8.01)	33.0 (±9.42)
	Lunetta et al. (2015) [15]	Cases	39.1 (±4.7)	37.0 (±5.1)	35.1 (±6.2)	32.8 (±6.5)
		Control	38.3 (±5.1)	38.1 (±4.3)	34.3 (±6.4)	28.7 (±7.5)

**Table 4 ijerph-18-01074-t004:** Summary of secondary outcome variables FVC, FSS, 6MWD.

Variable	Study		Baseline	End of Study		
				1 Month	3 Months	6 Months
FVC (%)	Clawson LL et al. (2017) [13]	Cases	88.14 (±11.03)	-	-	74.34
		Control	91.19 (±77.9)	-	-	88.59
	Lunetta et al. (2015) [15]	Cases	92.5 (±23.3)	-	-	75.8 (±23.6)
		Control	93.9 (±14.7)	-	-	66.5 (±26.9)
	Kato et al. (2018) [5]	Cases	69.05	-	-	72.2
	Van Groenestijn et al. (2019) [17]	Cases	86.9 (±20.2)	-	79.22	-
		Control	95.4 (±15.4)	-	81.88	-
	Zucchi et al. (2019) [10]	Cases	91.88 (±18.98)	-	-	66.24 (±44.96)
		Control	90.70 (±17.68)	-	-	77.91 (±31.82)
FSS (1–7)	Sivaramakrishnan et al. (2019) [16]	Cases	32.87 (±10.45) ÷ 9	28.62 (±11.9) ÷9	-	-
	Merico et al. (2018) [11]	Cases	5.4 (±0.27)	6.69 (±0.21)	-	
		Control	5.4 (±0.2)	5.19 (±0.16)	-	-
	Zucchi et al. (2019) [10]	Cases	35.63 (±15.31) ÷ 9	-	-	41.42 (±18.49) ÷ 9
		Control	36.50 (±16.53) ÷ 9	-	-	37.38 (±18.73) ÷ 9
6MWD (metros)	Sivaramakrishnan et al. (2019) [16]	Cases	232.5 (±192.32)	235.16 (±195.49)	-	-
	Merico et al. (2018) [11]	Cases	265.17 (±81.37)	336.73 (±50.72)	-	-
		Control	236.26 (±76.26)	239.16 (±5.48)	-	-

%FVC: percentage of forced vital capacity, FSS: fatigue severity scale, 6MWD: 6 min walk test.

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
