# Peer review of "Systematic Review of Therapeutic Physical Exercise in Patients with Amyotrophic Lateral Sclerosis over Time"

_ijerph, 2021, doi:10.3390/ijerph18031074_

Round 1
Reviewer 1 Report
As attached comments...

Author Response
International Journal of Environmental Research and Public Health
RESPONSE TO REVIEWER 1: Itemized List
(Manuscript ID: ijerph-1024917)
Title: The Effectiveness of Therapeutic Physical Exercise in Patients with Amyotrophic Lateral Sclerosis in Relation to the ALSFRS-R Functionality Scale: A Systematic Review
We would like to thank the Editor and Reviewers for their thoughtful and constructive comments. We have considered all suggestions, and have incorporated them into the revised manuscript. Changes to the original manuscript are identified by highlights (in yellow background). After corrections made, we believe that our document is much easier to read and understand. An itemized point-by-point response to the Reviewers’ comments is presented below.
Reviewer #1:
General comments:
Title
It does not represent what was done in the study, besides presenting abbreviations before presenting them in full, which is not usual in studies.
- Authors: Thank you very much for your comment. We have made changes to the title indicating the temporality of the effects to be evaluated of therapeutic physical exercise as an intervention strategy in patients with ALS. In addition, we have introduced the full development of the abbreviation.
Abstract
It does not represent what was done in the study, besides presenting abbreviations before presenting them in full, which is not usual in studies. It does not present the main results found. It does not present a relationship between the proposed title and the abstract as presented. It is difficult to list the title with the summary presented.
- Authors: Thank you very much for your comment. We have updated and rewritten a large part of the abstract. Indeed, it did not perfectly represent the contents of the article. We hope that after the review, this aspect has improved to facilitate the understanding of the article's contents to potential readers.
Introduction
Not from the general to the specific. It starts with an overview of the pathology, but then mentions the instrument and then moves on to physical therapy procedures. There is no link between the topics covered.
- Authors: Thank you very much for your comment. Indeed, there was no development of ideas aligned in the introduction. In this sense, the contents of the introduction have been rearranged and some paragraphs have been introduced that allow one idea to be linked to another to favor the development from the general to the specific.
The problem is not clear and is not presented in a robust way. It would be good to present study references to support the scientific gap.
- Authors: Thank you very much for your comment. In the introduction it has been shown that there is not enough evidence about what type of exercise as well as frequency or intensity would be necessary to treat patients with ALS. In addition, the importance of using a tool that allows comparing the results between the different interventions has been highlighted. Due to specificity and frequency of use, we consider that the appropriate tool is the “Revised Amyotrophic Lateral Sclerosis Functional Scale”.
The text suggests that the pathology treated with exercises will be evaluated, and the instrument to be used would be the ALSFRS-R. Thus, the instrument would not be the focus of the study and only a way to evaluate the intervention.
- Authors: Thank you very much for your comment. Indeed, that is the intention of the authors, to analyze the effects of therapeutic physical exercise using a common tool that allows the results to be compared. In this sense, we have modified the title to show that the ALSFRS-R is the instrument for evaluating the intervention.
Methods
The search criteria, such as search language, search period with month and year of start and end should be better defined.
- Authors: Thank you very much for your comment. Thank you very much for your comment. It is indeed very relevant information and had not been included in the document. It’s possible to find it in the material and methods section, specifically at the end of the “search strategy” and “selection of document” sections respectively.
Another fact that calls attention is that not all databases have the same scientific rigor. How was it done to solve this problem? Has a cut-off point been adopted in terms of the impact factor?
- Authors: Thank you very much for your comment. We absolutely agree that all databases do not have the same rigor. However, we think that in all databases it is possible to find a document that meets the inclusion / exclusion criteria. For this reason, the safe methodology was to analyze the identified documents and select those that reflected internal validity, according to the PEDro scale equal to or greater than 5 points.
Were all physical therapy exercises in the study? Apart from the PEDro and PRISMA scale, was any other scale adopted? For example PICOS method, or even some other form of quality control of studies.
- Authors: Thank you very much for your comment. No filter was made according to the type of exercise performed, since precisely the type of exercise as well as its intensity were the main questions that we wanted to answer with this document. On the other hand, the PICOT methodology (Patient; Intervention; Comparison; Outcome; Time) was followed. We have included precisely this information in the material and methods section.
Results
Are presented satisfactorily.
- Authors: Thank you very much for your comment.
Discussion
Despite being relatively well written, it does not refer to the numerical results of the studies used in the discussion. On the other hand, once again the ALSFRS-R instrument is used in a secondary way. Please explain this.
- Authors: Thank you very much for your comment. We have introduced the most significant values of the changes experienced by the participants in the short, medium and long term in the main outcome variable
Conclusion
The study has its conclusions focused on therapies and not on the evaluation of these therapies.
- Authors: Thank you very much for your comment. It is precisely the idea that we pursued, to identify the effectiveness of the intervention, using a common outcome measure as a strategy to be able to compare the results. In this sense, we have not made any changes to the conclusion, but rather we have made changes to the document to precisely identify the ALSFRS-R as an instrument that allows comparison between the different interventions. However, if the reviewer feels that additional changes should be made to this section, please do not hesitate to let us know.
References
Are presented satisfactorily, however the format used is not the one determined by the journal.
- Authors: Thank you very much for your comment. We have reviewed the references according to the format of the journal and we have made the pertinent changes.
Overview
The manuscript presented addresses a relevant research topic.
It would be advisable to do a general review.
Specific comments and suggestions:.
- Authors: Thank you very much for the time spent reviewing this document. After making the changes suggested by the reviewer, we consider that the document has improved a lot structurally and it is much easier to understand the idea. In addition, we have sent the document to a professional proofreader to correct any grammar / vocabulary errors. However, if the reviewer considers that one of his comments has not been answered correctly, we will let him know so we can address it correctly.
What do you want to do ? New mailCo

Reviewer 2 Report
The work of Ortega-Hombrados and co-authors systematically reviewed the role of physical therapy and exercise in ALS patients. The paper require refinements to be suitable for publications. Minor and Major concerns are reported below. This reviewer suggests minor revision for the work.
ABSTRACT
The abstract should be improved, I suggest to remove the Background-Methods-Results-and Discussions structure in the abstract. Please consider to rewrite this section in a fluent way without over-structures. However this constitute a minor concern. I mainly suggest to carefully rewrite the Background and the conclusions.
INTORDUCTION
The introduction presents some minor concerns regarding the written English. And one major concern.
As first in the statement :"The main components of this ....ALSFRS-R scale [5]." Please remove the underlines. However the main problem is the last part of the section: "Everything indicates ... 6-minute walk test (6MWT)." Here there are repetitions such as "In addition". Moreover, it does not emerge the aim of the review paper you proposed. The aim you write seems much more the aim of a research paper and thus it is not coherent with the entire work. I suggest to rewrite this section, highlighting the necessity of a review paper that treats physical therapy in ALS patients.
METHODS
Please consider to refer to the PRISMA model (figure 1) the first time that appear in the text, if possible explain PRISMA. Add a caption to figure 1.
RESULTS
Few minor concerns, tables are not fully readable. Manage the pages in order to provide fully readable and appropriate table from a graphical point of view. Moreover add captions to each single table.
DISCUSSION and CONCLUSION
Both sections are coherent with the results of the review performed. Authors used the expression "Other studies..." but they eventually cite just one paper to support the statement. I suggest to rephrase.
Author Response
International Journal of Environmental Research and Public Health
RESPONSE TO REVIEWER 2: Itemized List
(Manuscript ID: ijerph-1024917)
Title: The Effectiveness of Therapeutic Physical Exercise in Patients with Amyotrophic Lateral Sclerosis in Relation to the ALSFRS-R Functionality Scale: A Systematic Review
We would like to thank the Editor and Reviewers for their thoughtful and constructive comments. We have considered all suggestions, and have incorporated them into the revised manuscript. Changes to the original manuscript are identified by highlights (in yellow background). After corrections made, we believe that our document is much easier to read and understand. An itemized point-by-point response to the Reviewers’ comments is presented below.
Reviewer #2:
The work of Ortega-Hombrados and co-authors systematically reviewed the role of physical therapy and exercise in ALS patients. The paper require refinements to be suitable for publications. Minor and Major concerns are reported below. This reviewer suggests minor revision for the work.
- Authors: Thank you very much for your comment and for the time spent reviewing and correcting the document. The authors agree that, after the changes made, the document has improved structurally much and is easier to understand for potential readers.
ABSTRACT
The abstract should be improved, I suggest to remove the Background-Methods-Results-and Discussions structure in the abstract. Please consider to rewrite this section in a fluent way without over-structures. However this constitute a minor concern. I mainly suggest to carefully rewrite the Background and the conclusions.
- Authors: Thank you very much for your comment.
INTRODUCTION
The introduction presents some minor concerns regarding the written English. And one major concern. As first in the statement:"The main components of this ....ALSFRS-R scale [5]." Please remove the underlines. However the main problem is the last part of the section: "Everything indicates ... 6-minute walk test (6MWT)." Here there are repetitions such as "In addition". Moreover, it does not emerge the aim of the review paper you proposed. The aim you write seems much more the aim of a research paper and thus it is not coherent with the entire work. I suggest to rewrite this section, highlighting the necessity of a review paper that treats physical therapy in ALS patients.
- Authors: Thank you very much for your comment and suggestion. We have modified the introduction section removing the underlines, rewriting important parts of this section highlighting the necessity of a review paper where the physical therapy in ALS patients was analysed.
METHODS
Please consider to refer to the PRISMA model (figure 1) the first time that appear in the text, if possible explain PRISMA. Add a caption to figure 1.
- Authors: Thank you very much for your suggestion. We have added an explanation about PRISMA acronym and a caption to figure 1 the first time that it is mentioned in the document.
RESULTS
Few minor concerns, tables are not fully readable. Manage the pages in order to provide fully readable and appropriate table from a graphical point of view. Moreover add captions to each single table.
- Authors: Thank you very much for your comment. We have managed the pages in order to provide fully readable table. We must to edit the table 2 to make it appropriate from a graphical point of view. In addition, we have added captions to each table.
DISCUSSION and CONCLUSION
Both sections are coherent with the results of the review performed. Authors used the expression "Other studies..." but they eventually cite just one paper to support the statement. I suggest to rephrase.
- Authors: Thank you very much for your comment. We have revised and modified these wrong expressions.

Reviewer 3 Report
Great article! , very informative. Clinically relevant for healthcare professionals across the globe. Grammatical errors exist in the sentence structure. The article needs professional editing.
Writers need to send the article for professional editing. There are few if not moderate grammatical errors. The grammar errors are beyond the scope of the writers to correct. There has been an omission of word modifiers, conjunctions, and other over usage of adjectives.
Author Response
International Journal of Environmental Research and Public Health
RESPONSE TO REVIEWER 3: Itemized List
(Manuscript ID: ijerph-1024917)
Title: The Effectiveness of Therapeutic Physical Exercise in Patients with Amyotrophic Lateral Sclerosis in Relation to the ALSFRS-R Functionality Scale: A Systematic Review
We would like to thank the Editor and Reviewers for their thoughtful and constructive comments. We have considered all suggestions, and have incorporated them into the revised manuscript. Changes to the original manuscript are identified by highlights (in yellow background). After corrections made, we believe that our document is much easier to read and understand. An itemized point-by-point response to the Reviewers’ comments is presented below.
Reviewer #3:
Great article! , very informative. Clinically relevant for healthcare professionals across the globe. Grammatical errors exist in the sentence structure. The article needs professional editing.
- Authors: Thank you very much for your comments and for the time invested in correcting it. Undoubtedly, your suggestion of sending it to a professional proof-reader has improved the grammar and vocabulary used and, therefore, favors the understanding of the text for potential readers.
Writers need to send the article for professional editing. There are few if not moderate grammatical errors. The grammar errors are beyond the scope of the writers to correct. There has been an omission of word modifiers, conjunctions, and other over usage of adjectives..
- Authors: Thank you very much for your suggestion. We have sent the document to be corrected by a professional proof-reader.

Round 2
Reviewer 1 Report
The only consideration I have is that the title was supposed to only have about 14 words. The rest is well to me.
Author Response
Dear Reviewer,
thanks again for the time invested in reviewing the document again. We have rewritten the title adjusting it to the number of words indicated, giving preference to four aspects: type of document, intervention, patient profile and evaluation over time. Please, if you think that we should make eventual modifications, do not hesitate to indicate it.
Grateful.
What do you want to do ? New mailCopy
